# Analysis of disease burden and future trends of ischemic heart disease in China and globally, 1990–2023

Zhuo Zhang[1], Guihua Yue[1]*, Longlong Wang [2]*

**1** Guangxi University of Chinese Medicine, Nanning, China, **2** International Zhuang Medicine Hospital Affiliated to Guangxi University of Chinese Medicine, Nanning, China

* 54960313@qq.com (GY); wanglonglong2022@stu.gxtcmu.edu.cn (LW)

## Abstract

### Background

This study aimed to analyze the characteristics and trends of the disease burden of ischemic heart disease (IHD) in China and worldwide, across different age and sex groups, to provide a comprehensive understanding of the situation in China.

### Methods

Using GBD 2023 data, we analyzed global and China-specific IHD incidence, prevalence, mortality, and DALYs. Joinpoint regression was used to assess trends, while factors such as aging and population growth were also examined. An ARIMA model projected the IHD burden from 2024 to 2050.

### Results

From 1990 to 2023, the age-standardized incidence rate (ASIR) and prevalence rate (ASPR) of IHD in China exhibited an upward trend. The ASIR per 100,000 increased from 109.7 (95% uncertainty interval [UI]: 93–128) to 110.3 (95% UI: 89–131), while the age-standardized mortality rate (ASMR) decreased from 126.8 (95% UI: 92–163) to 109.5 (95% UI: 94–127), mirroring global trends. The highest incidence of IHD was observed in individuals aged 60–89 years, with mortality peaking between 80–89 years. AAPC analysis revealed that China's ASIR (AAPC = −0.091%; 95% CI: −0.152 to −0.029) and ASPR (AAPC = 0.227%; 95% CI: 0.115 to 0.34) increased more rapidly than the global average. Decomposition analysis indicated that the rise in IHD burden in China was primarily driven by population aging and growth, a pattern observed globally. Forecasting models suggest that the ASIR will continue to rise in China through 2050, while the ASMR and age-standardized DALYs rates (ASDR) are expected to decline further.

**Data availability statement:** All data generated or analysed during this study are included in this published article or its Supporting Information files.

**Funding:** We confirm that this funding project is: Guangxi Excellent Scholar Training Project (GXQH202403).

**Competing interests:** The authors have declared that no competing interests exist.

## Conclusion

The incidence and prevalence of IHD in China are projected to rise through approximately 2030. Older males bear a disproportionate burden of the disease. More targeted preventive, diagnostic, and therapeutic strategies are urgently needed and should be informed by the findings of this study.

## Introduction

Ischemic heart disease (IHD) is one of the most prevalent cardiovascular conditions worldwide [1], characterized by very high incidence and mortality rates. It poses a significant global threat to public health [2] and represents a major challenge to the health systems [3]. The incidence and mortality of IHD are heavily influenced by social factors, including environmental pollution, dietary habits, occupational stress, and mechanization [4,5]. Additionally, they are closely linked to geographic factors, including genetics, altitude, and climate [6], reflecting the strong geographical variability of the disease. Although IHD is most prevalent among older adults, recent studies and clinical observations suggest that the onset of IHD is occurring at increasingly younger ages [7]. Therefore, it is particularly important to clearly present the epidemiological characteristics of IHD, the associated disease burden on both individuals and society, and its developmental trajectory. These elements form the foundation for implementing long-term disease control initiatives. They are crucial for accurately identifying key areas for prevention and intervention in IHD [8], thereby supporting the development of effective prevention and control strategies.

The Global Burden of Disease (GBD) study is a large-scale observational epidemiological project that covers all regions worldwide. While previous studies have utilized GBD 2019 and 2021 data to analyze the IHD burden in China and globally, the emergence of a younger demographic affected by IHD, as well as the availability of updated data from 2022 and 2023, have not yet been incorporated into existing analyses. Consequently, there is a pressing need for a more current and comprehensive evaluation of IHD using the updated GBD database, which better reflects recent trends and provides more representative data.

Considering the above, this study compiled and analyzed the statistical data from the most recent GBD 2023 database, employing advanced and comprehensive statistical methods to conduct a comparative analysis of the burden and risk factors of IHD and their trends in China and globally from 1990 to 2023. Furthermore, the study forecasted the future trajectory of IHD burden in China and globally, providing robust data support for enhancing cardiovascular disease prevention strategies, optimizing healthcare services, and developing informed health policies.

## Materials and methods

### Data sources

The data for this study were obtained from the Global Burden of Disease (GBD) 2023 database. Compared to the 2021 version, this updated database includes more

detailed records on the incidence, prevalence, mortality, and disability-adjusted life years (DALYs) for over 300 diseases across 206 countries and regions [9,10], enabling comprehensive research on disease epidemiology, burden, and trends. We extracted data on the incidence, prevalence, mortality, and DALYs related to ischemic heart disease (IHD), along with their corresponding 95% uncertainty intervals (UIs), in China and globally, stratified by sex and age group, from 1990 to 2023. No special ethical approval was required, as all data were sourced from a publicly available database. This study adhered to rigorous and transparent reporting guidelines for health assessments.

## Statistical analysis

All statistical analyses and data visualization were conducted using R software (version 4.4.3) and the following R packages: "readxl," "dplyr," "ggplot2," "reshape2," and "forecast." These packages were used for data cleaning, statistical computation, visualization, and prediction of future trends. A p-value less than 0.05 was considered statistically significant.

## Joinpoint analysis

Joinpoint Regression Program version 5.2.0.0 was used to calculate the annual percentage change (APC), average annual percentage change (AAPC), and their corresponding 95% confidence intervals (CIs) for IHD incidence, prevalence, mortality, and DALYs in China and globally from 1990 to 2023. The best-fitting model was selected for each time series to assess temporal trends in disease burden. An AAPC with a 95% CI greater than 0 indicates an increasing trend; less than 0 indicates a decreasing trend; and equal to 0 suggests a stable trend.

## Decomposition analysis

We employed the decomposition method developed by Das Gupta to assess the effects of population growth, population aging, and epidemiological changes in IHD burden. This classical approach disaggregates total changes into specific components, allowing for the quantification of individual contributions. Using this method, we examined how changes in age structure, population size, and age-adjusted epidemiological factors influenced the burden of IHD in China and globally. Epidemiological changes refer to morbidity and mortality rates adjusted for the baseline population and age distribution.

## Model prediction

The Autoregressive Integrated Moving Average (ARIMA) model is a widely used technique in time series analysis. It combines three components, autoregressive (AR), differencing (I), and moving average (MA), to effectively capture trends and cyclical patterns in data and predict future changes. In building the ARIMA (p, d, q) model, 'p' represents the number of autoregressive terms, 'd' the order of differencing, and 'q' the number of moving average terms. First, we applied differencing to stabilize the time series and used the Kwiatkowski–Phillips–Schmidt–Shin (KPSS) test to assess stationarity. Q–Q plots were then used to evaluate whether the residuals followed a normal distribution. We employed the auto.arima() function to automatically select the optimal model based on the lowest Akaike Information Criterion (AIC) and Bayesian Information Criterion (BIC) values [11]. Finally, the Ljung–Box test was performed to evaluate the randomness of residuals. When residuals exhibit characteristics of white noise, the ARIMA model is considered a reliable linear predictor for short-term forecasting.

## Sociodemographic index

The Sociodemographic Index (SDI) is a composite indicator that reflects the level of social and demographic development of a country or region. It enables comparison of the relationship between socioeconomic development and health burden across different settings. The SDI is scored on a scale from 0 to 1, where 0 indicates the lowest levels of per capita income, educational attainment, and the highest total fertility rate among all GBD locations. Conversely, a score of 1 reflects the highest income, highest educational levels, and lowest fertility rate.

## Disability-adjusted life years

Disability-adjusted life years (DALYs) are a widely accepted composite health metric used to quantify the loss of healthy life due to illness, injury, or premature death. DALYs are a core outcome measure in the Global Burden of Disease study and serve as a critical indicator for health policymakers. They provide essential insights into the overall impact of diseases, health conditions, or interventions on population health.

## Results

### Basic description of the burden of disease for IHD in China and globally

As shown in Table 1, from 1990 to 2023, both the age-standardized incidence rate (ASIR) and age-standardized prevalence rate (ASPR) of ischemic heart disease (IHD) showed an upward trend in the total population of China. The ASIR increased slightly from 109.7 cases per 100,000 population (95% uncertainty interval [UI]: 93–128) to 110.3 cases per 100,000 (95% UI: 89–131). In contrast, the global ASIR decreased significantly from 285.1 cases per 100,000 (95% UI: 252–320) to 177.7 cases per 100,000 (95% UI: 154–203). Similarly, China's ASPR rose from 3,259 cases per 100,000 (95% UI: 2,735–3,931) to 3,448.8 cases per 100,000 (95% UI: 2,923–4,101), while the global ASPR declined from 4,187.9 cases per 100,000 (95% UI: 3,688–4,728) to 3,204.5 cases per 100,000 (95% UI: 2,832–3,621). The age-standardized mortality rate (ASMR) in China decreased from 126.8 per 100,000 (95% UI: 92–163) to 109.5 per 100,000 (95% UI: 94–127), and globally from 190.3 per 100,000 (95% UI: 174–208) to 124.9 per 100,000 (95% UI: 113–138). Similarly, the age-standardized DALYs rate (ASDR) declined in China from 2,641.2 per 100,000 (95% UI: 1,913–3,382) to 2,133.3 per 100,000 (95% UI: 1,869–2,420), and globally from 4,081.8 per 100,000 (95% UI: 3,699–4,549) to 2,790.5 per 100,000 (95% UI: 2,517–3,096). Between 1990 and 2023, the average annual percentage changes (AAPCs) for China were: ASIR: –0.091% (95% confidence interval [CI]: –0.152 to –0.029), ASPR: 0.227% (95% CI: 0.115 to 0.34), ASMR: –0.422% (95% CI: –0.744 to –0.1), ASDR: –0.810% (95% CI: –1.088 to –0.531). In comparison, global AAPCs were: ASIR: –1.556% (95% CI: –1.634 to –1.477), ASPR: –0.921% (95% CI: –1.016 to –0.827), ASMR: –1.411% (95% CI: –1.662 to –1.159), ASDR: –1.248% (95% CI: –1.5276 to –0.9679). Overall, while ASIR and ASPR have been increasing in China, ASMR and ASDR have been declining— a trend that contrasts with the global pattern, where all four indicators have decreased over

**Table 1. Burden of Disease for IHD in China and Globally in males and females.**

| Location | Measure | 1990 | | 2023 | | 1990-2023 AAPC |
|---|---|---|---|---|---|---|
| | | All-ages cases | Age-Standard rate per 100,000 people | All-ages cases | Age-Standard rate per 100,000 people | |
| China | Incidence | 522373 (448905-616511) | 109.7 (93-128) | 1906927.5 (1568663-2268303) | 110.3 (89-131) | −0.091 (−0.152 - −0.029) |
| | Prevalence | 24257685.9 (20534742-29620524) | 3259 (2735-3931) | 66695396.7 (55561821-79427078) | 3448.8 (2923-4101) | 0.227 (0.115 - 0.34) |
| | Deaths | 727798.3 (579668-866725) | 126.8 (92-163) | 1982120.4 (1705599-2246616) | 109.5 (94-127) | −0.422 (−0.744 - −0.1) |
| | DALYs | 18509970.2 (14623754-22380627) | 2641.2 (1913-3382) | 36913769.7 (32438691-41193393) | 2133.3 (1869-2420) | −0.8098 (−1.088 - −0.5309) |
| Global | Incidence | 8450216.9 (7530917-9593255) | 285.1 (252-320) | 12731128.3 (11077432-14673673) | 177.7 (154-203) | −1.556 (−1.634 - −1.477) |
| | Prevalence | 143363043.8 (126429458-164157947) | 4187.9 (3688-4728) | 239538555.3 (210988752-272406233) | 3204.5 (2832-3621) | −0.921 (−1.016 - −0.827) |
| | Deaths | 5540571.6 (5116276-5940280) | 190.3 (174-208) | 8942485 (8059693-9686404) | 124.9 (113-138) | −1.411 (−1.662 - −1.159) |
| | DALYs | 127007145.2 (116101217-137105687) | 4081.8 (3699-4549) | 193838763.9 (176085091-211101683) | 2790.5 (2517-3096) | −1.2481 (−1.5276 - −0.9679) |

time. As detailed in Tables 2 and 3, from 1990 to 2023, the ASPR increased in both male and female populations in China, while ASIR, ASMR, and ASDR followed the global decreasing trend in both sexes.

Fig 1 illustrates changes in ASIR, ASPR, ASMR, and ASDR for the total population, as well as by age group and sex, in China and globally from 1990 to 2023. Across all age groups, these indicators first increased and then declined with age. Compared with 1990, there was a notable increase in values across all age groups by 2023. In 2023, the age of onset for IHD in both China and globally was primarily concentrated after age 40. Notably, ASIR, ASPR, ASMR,

**Table 2. Burden of Disease for IHD in China and Globally in males.**

| Location | Measure | 1990 | | 2023 | | 1990-2023 AAPC |
|---|---|---|---|---|---|---|
| | | All-ages cases | Age-Standard rate per 100,000 people | All-ages cases | Age-Standard rate per 100,000 people | |
| China | Incidence | 270378.4 (230603-322931) | 92.3 (78-107) | 1052996.5 (864469-1248641) | 89.5 (73-106) | 0.198 (−0.091 - 0.487) |
| | Prevalence | 13840386.6 (11680569-16921839) | 2813.7 (2366-3383) | 36449627.4 (30662385-43704685) | 3020 (2553-3605) | 0.227 (0.115 - 0.34) |
| | Deaths | 383105.5 (276955-494796) | 112.4 (92-132) | 1066173.4 (926923-1238037) | 92.7 (79-105) | −0.317 (−0.629 - −0.005) |
| | DALYs | 10510831.1 (7485694-13603860) | 2282.2 (1837-2713) | 22084874.6 (19279459-25075601) | 1688.1 (1484-1879) | −0.549 (−0.862 - −0.235) |
| Global | Incidence | 4513364.8 (4005137-5162831) | 236.3 (211-264) | 7305238.8 (6290689-8372914) | 140.8 (122-161) | −0.829 (−0.92 - −0.737) |
| | Prevalence | 79402526.8 (70048091-90718335) | 3544.5 (3120-4027) | 137406562.2 (120924020-156173779) | 2636 (2328-2996) | −0.921 (−1.016 - −0.827) |
| | Deaths | 2891641 (2625158-3201306) | 161.3 (147-173) | 4995565.1 (4535025-5532336) | 100 (90-108) | −1.275 (−1.556 - −0.994) |
| | DALYs | 74122456.9 (66577537-83530670) | 3274.7 (3021-3529) | 118773044.4 (106793634-131749523) | 2141.1 (1942-2331) | −1.097 (−1.387 - −0.806) |

**Table 3. Burden of Disease for IHD in China and Globally in females.**

| Location | Measure | 1990 | | 2023 | | 1990-2023 AAPC |
|---|---|---|---|---|---|---|
| | | All-ages cases | Age-Standard rate per 100,000 people | All-ages cases | Age-Standard rate per 100,000 people | |
| China | Incidence | 251994.6 (214745.4-302221.6) | 80.5 (68-95.4) | 853931.1 (691750.6-1030315.7) | 72.5 (58.8-87.2) | 0.215 (0.187 - 0.243) |
| | Prevalence | 10417299.3 (8851514.8-12731152.6) | 2398.5 (2009.3-2869) | 30245769.3 (24941786.7-36415606.8) | 2581.7 (2165.4-3112.8) | 0.227 (0.115 - 0.34) |
| | Deaths | 344692.8 (247040.8-427571.9) | 100.2 (73-124.3) | 915947 (728048.9-1118203.8) | 77.2 (60.9-94.3) | −0.743 (−1.115 - −0.37) |
| | DALYs | 7999139.1 (5704544.8-10335461) | 1943.8 (1406.2-2454.4) | 14828895.1 (11991234.5-17631054.9) | 1254.3 (1021.1-1485.6) | −1.292 (−1.645 - −0.938) |
| Global | Incidence | 3936852.2 (3507321.3-4432010.5) | 196.5 (175.4-219.5) | 5425889.5 (4748427.2-6189745.7) | 109.4 (95.9-124.6) | −1.029 (−1.074 - −0.985) |
| | Prevalence | 63960517 (56519511-73454879.3) | 2964 (2602.6-3389.4) | 102131993.1 (89749337.1-117143088.8) | 2116.3 (1863-2426.3) | −0.921 (−1.016 - −0.827) |
| | Deaths | 2648930.6 (2363041.4-2918803.9) | 136.3 (121-149.2) | 3946919.9 (3339170.6-4457322.1) | 78.3 (66.3-88.5) | −1.629 (−1.851 - −1.407) |
| | DALYs | 52884688.3 (46645175.1-59580104.5) | 2534.2 (2235.7-2832.9) | 75065719.5 (63724571.8-86844712.6) | 1539.6 (1305-1784.4) | −1.47 (−1.613 - −1.327) |

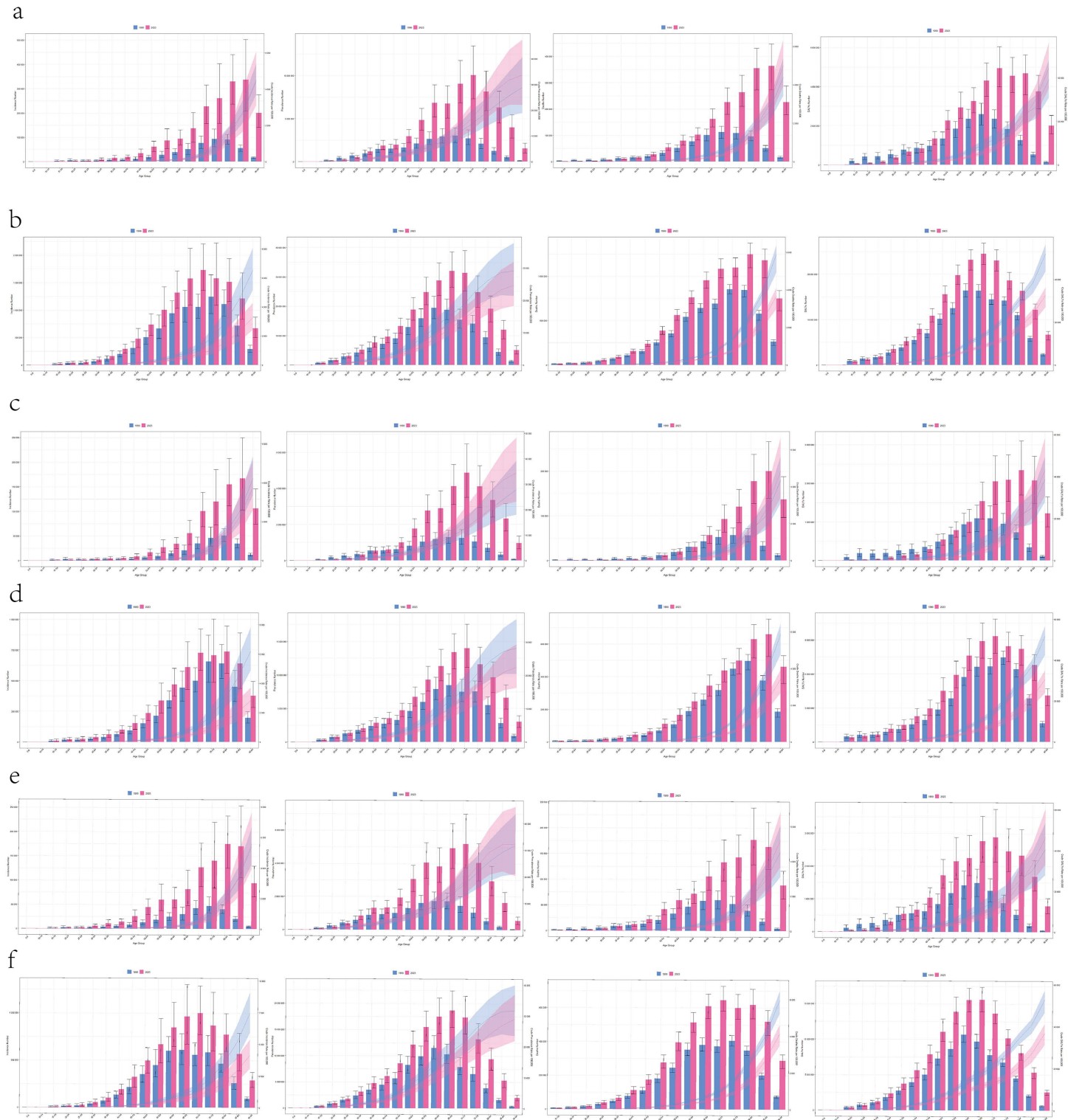

**Fig 1. Comparison of incidence rates, prevalence rates, mortality rates, and DALYs rates by age group and their crude rates from 1990 to 2023.** (a: The total population of IHD in China. b: The total population of IHD in Globally. c: female IHD in China. d: female IHD in Globally. e: male IHD in China. f: male IHD in Globally). The black line and shaded area represent the 95% UI. DALYs, disability-adjusted life years; UI, uncertainty interval.

and ASDR values in China for individuals under 29 years of age were significantly lower than in 1990. The global peak in burden appeared earlier than in China, with similar patterns observed in both male and female groups. From 1990 to 2023, both the number of IHD-related deaths and the DALY rate increased across all age groups. The majority of deaths were concentrated among those aged 65–89 years, with the highest proportion occurring in the 80–89-year-old group in China and the 80–84-year-old group globally. Fig 2 shows the relationship between ASIR and ASMR and the Sociodemographic Index (SDI). ASIR initially increases with SDI in both China and the global population, but then declines as SDI continues to rise. Although China's ASDR for both sexes fluctuates with increasing SDI, it generally shows a declining trend. Notably, China's ASIR remains significantly lower than expected when compared to other countries with a similar SDI level.

## Joinpoint regression analysis of heart failure disease burden in China and globally

Joinpoint regression analyses of the age-standardized incidence rate (ASIR), prevalence rate (ASPR), mortality rate (ASMR), and disability-adjusted life years rate (ASDR) for the total population in China and globally from 1990 to 2023 are presented in Fig 3. The overall trend of IHD burden in China reveals a wave-like pattern. Specifically, ASIR exhibited an upward trend from 1990 to 2005, peaking in 2005, followed by a downward trend through 2023. ASPR increased significantly from 1990 to 2000, declined between 2000 and 2015, and then rose again from 2015 to 2023. ASMR demonstrated a significant rise from 1990 to 2005, followed by a decline. Between 2005 and 2012, the trend flattened and then slightly increased, before rising more markedly from 2012 to 2017. A substantial decline occurred from 2017 to 2020, followed by another increase from 2020 to 2023. ASDR followed a similar pattern, with a steady decline from 1990 to 2012, a

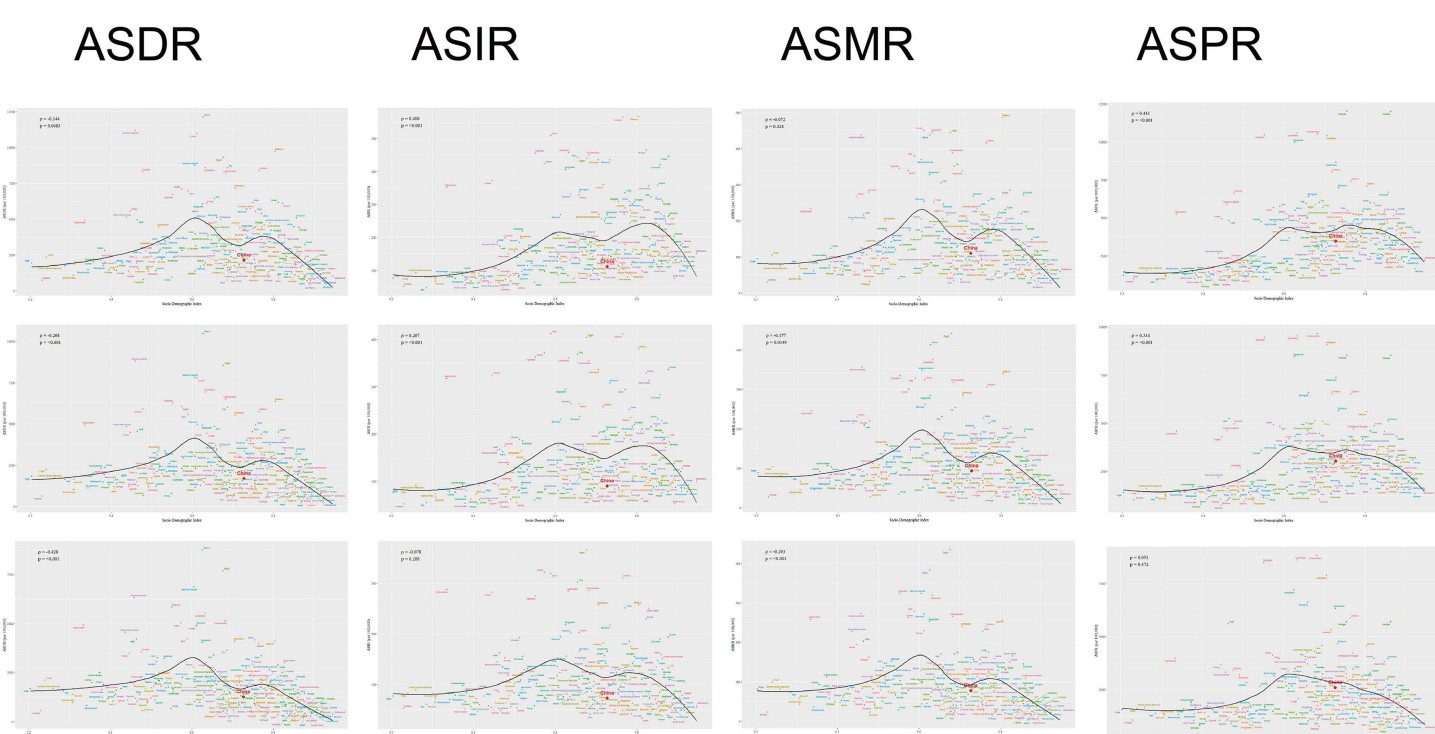

**Fig 2. Age-standardized burden rate attributable to IHD across 204 countries and regions by socio_x0002_demographic index, 1990–2023.** The black line was an adaptive association fitted with adaptive Loess regression based on all data points.

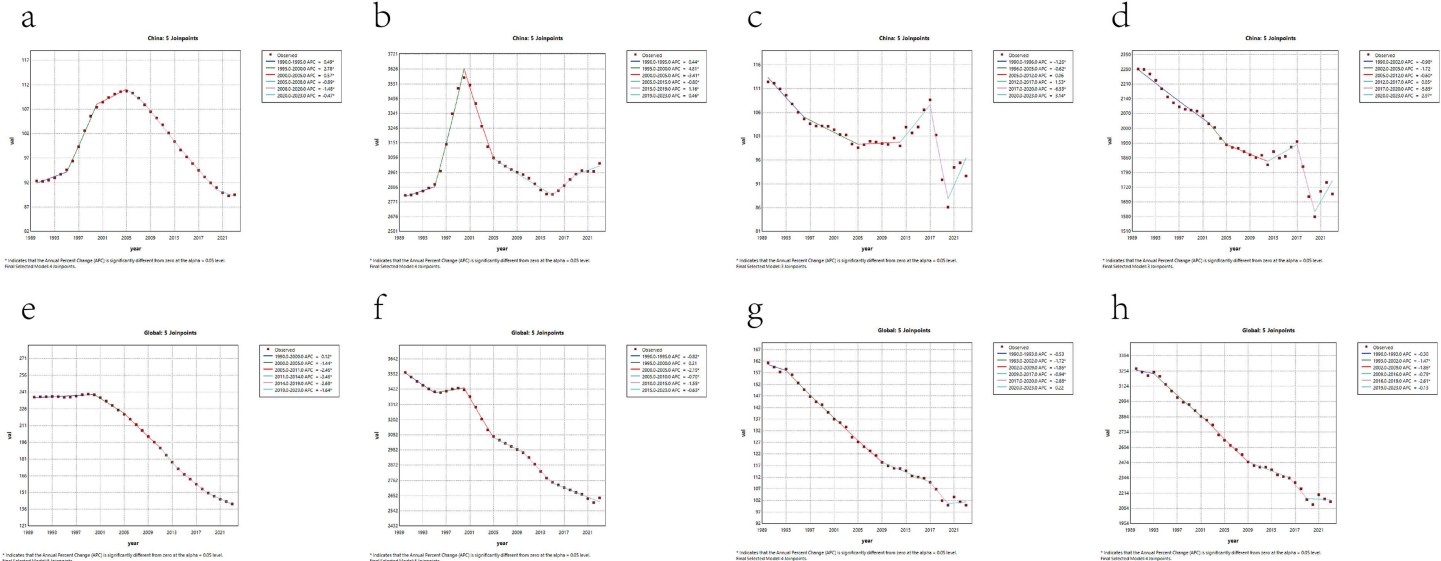

**Fig 3. The APC of ASIR, ASPR, ASMR, and ASDR in the total population of IHD in China.** (a-d) and globally (e-h) from 1990 to 2023 (* indicates P-value<0.05, statistically significant). a, e: ASIR; b, f: ASPR; c, g: ASMR; d, h: ASDR. APC, annual percent change; ASPR, age-standardized prevalence rate; ASDR, age-standardized DALYs rate.

moderate increase between 2012 and 2017, a sharp decline from 2017 to 2020, and another significant rise from 2020 to 2023. In contrast, the overall global trend in IHD burden showed a general decline. However, global ASPR increased during the period from 1995 to 2000. Additionally, both ASMR and ASDR exhibited a slow upward trend between 2020 and 2023, a development that warrants further attention and investigation.

Fig 4 displays Joinpoint regression analyses for the female populations in China and globally. Notably, the ASPR among Chinese females deviates from the pattern observed in the overall Chinese population. It exhibited a slight upward trend from 1990 to 1998, followed by a decline from 1998 to 2010, and then a continuous and significant increase from 2010 to 2023. Meanwhile, the trends in ASIR, ASMR, and ASDR in Chinese females closely resemble those observed in the total Chinese population. For the global female population, the trends for all four indicators are consistent with those observed in the global total population.

Fig 5 presents Joinpoint regression results for the Chinese and global male populations. The trends in ASIR, ASPR, ASMR, and ASDR among males are consistent with those observed in the total population, both in China and globally.

## Decomposition analysis

We analyzed the impact of population growth, population aging, and epidemiological changes on the IHD burden. As shown in Fig 6, decomposition analysis from 1990 to 2023 revealed that population aging and population growth significantly contributed to the increasing prevalence of IHD in both China and globally. These demographic shifts also led to higher mortality and DALYs rates. In China, changes in epidemiological trends—specifically, age-specific morbidity and mortality—also contributed to the increase in IHD prevalence. However, on a global scale, the rise in prevalence was not influenced by epidemiological changes, suggesting a fundamentally different driving mechanism compared to China. This pattern was consistent across both male and female populations.

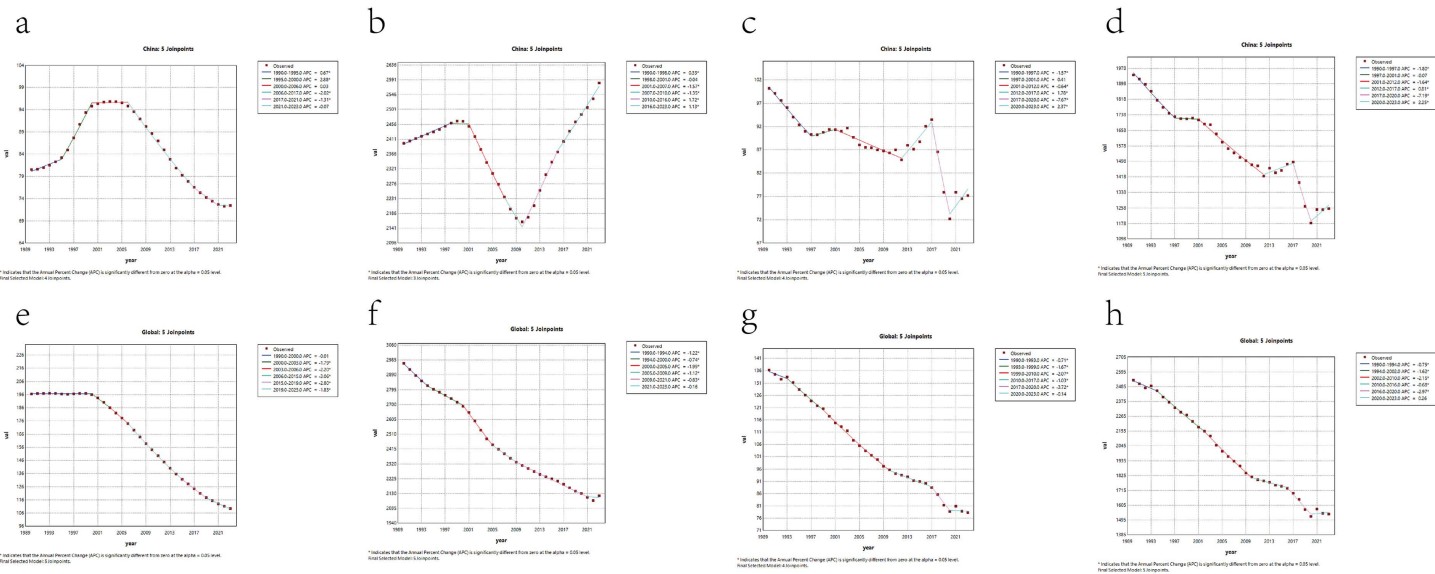

**Fig 4. The APC of ASIR, ASPR, ASMR, and ASDR in female IHD in China.** (a-d) and globally (e-h) from 1990 to 2023 (* indicates P-value<0.05, statistically significant). a, e: ASIR; b, f: ASPR; c, g: ASMR; d, h: ASDR.

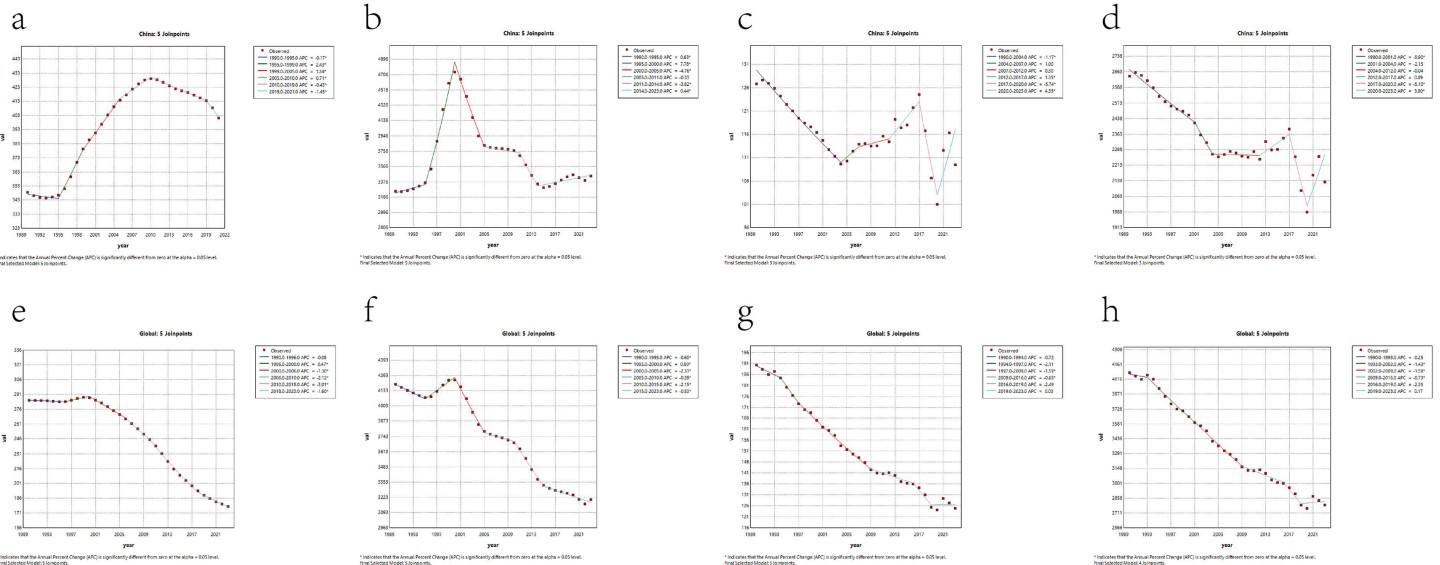

**Fig 5. The APC of ASIR, ASPR, ASMR, and ASDR in male IHD in China.** (a-d) and globally (e-h) from 1990 to 2023 (* indicates P-value<0.05, statistically significant). a, e: ASIR; b, f: ASPR; c, g: ASMR; d, h: ASDR.

## Trend forecast for IHD from 2024 to 2050

Fig 7 presents the projected trends of ischemic heart disease (IHD) in China and globally from 2024 to 2050. Table 4 confirms that the residuals of the ARIMA models exhibit white noise characteristics, as validated by the Ljung–Box test, indicating good model fit. According to ARIMA model projections, the ASIR in China's total population is expected to continue

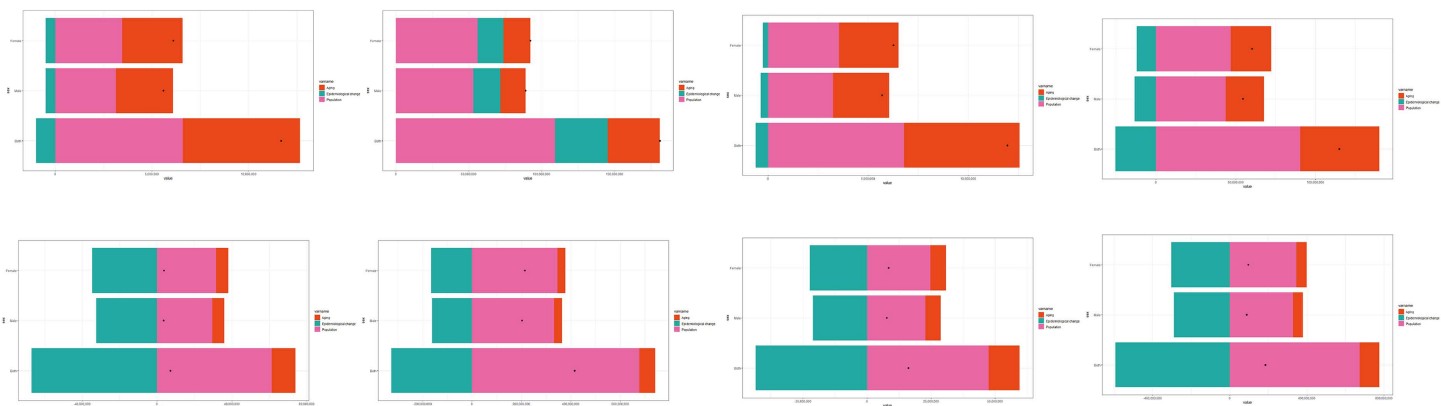

**Fig 6. Decomposition analysis of IHD in China and globally from 1990 to 2023.** Black dots represent the overall change values due to population growth, aging, and epidemiological changes.

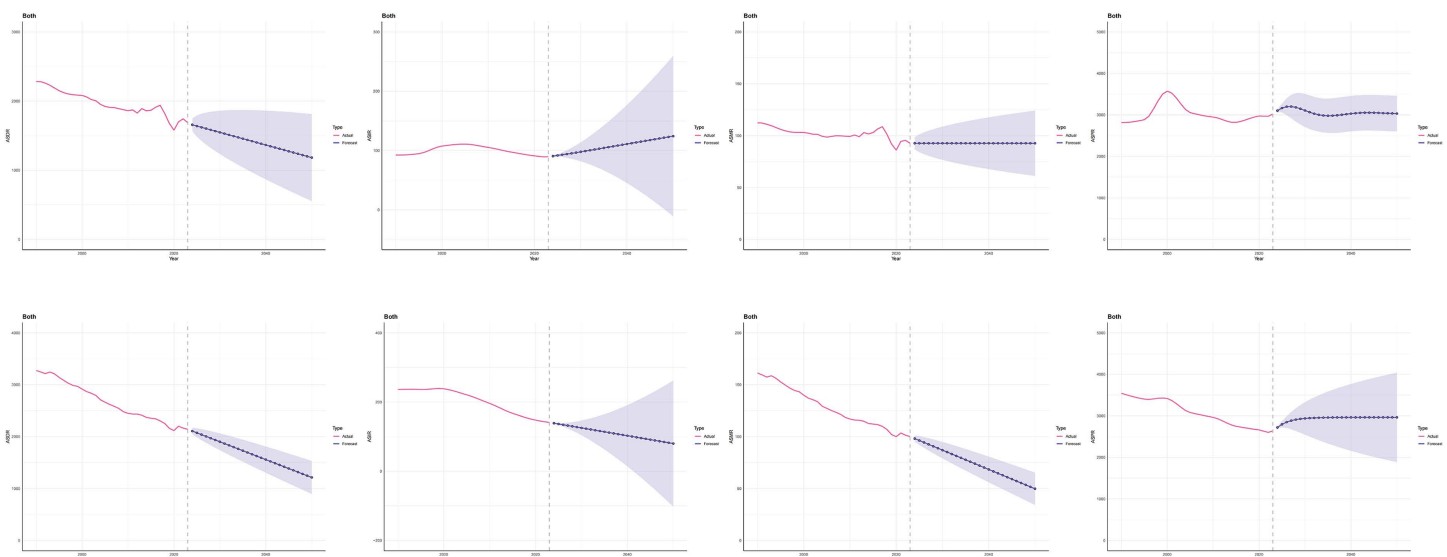

**Fig 7. Time trends of ASIR, ASPR, ASMR, ASDR in the total population of IHD in China from 1990 to 2050. blue dot lines and shaded regions represent the predicted trend and its 95% CI.**

rising, increasing from 89.5 to 124.2 cases per 100,000 people between 2023 and 2050. Similarly, the ASPR is expected to continue increasing, from 3,020.0 to 3,031.8 cases per 100,000, with a peak projected in 2027 at 3,201.3 cases per 100,000. In contrast, the ASMR is projected to remain stable, while the ASDR is expected to decline steadily, from 1,688.1 to 1,182.9 cases per 100,000 people by 2050. Globally, ASIR, ASMR, and ASDR are all projected to decline during this period. ASIR is forecasted to decrease from 140.8 to 80.0 cases per 100,000 people, while ASMR will decrease from 100.0 to 49.8 cases per 100,000, and ASDR will decrease from 2,141.1 to 1,213.6 cases per 100,000. The global ASPR, however, is expected to increase slightly before stabilizing, rising from 2,636.0 to 2,963.4 cases per 100,000.

Fig 8 shows the projections for the Chinese female population through 2050. ASIR is expected to rise gradually, from 72.5 to 83.1 cases per 100,000 people. The ASPR shows a pronounced rise followed by a decline, increasing from

**Table 4. Trend Forecast for IHD from 2024 to 2050.**

| Location | Measure | Sex | Model(p,d,q) | AIC | BIC | Ljung-Box(P) |
|---|---|---|---|---|---|---|
| China | Incidence | Both | (4,0,0) | 33.817 | 42.975 | 0.842 |
| | | Female | (2,2,0) | 10.86 | 15.258 | 0.731 |
| | | Male | (3,2,0) | 16.698 | 22.561 | 0.93 |
| | Prevalence | Both | (3,0,1) | 352.27 | 361.428 | 0.978 |
| | | Female | (2,0,2) | 222.909 | 232.067 | 0.606 |
| | | Male | (2,0,2) | 304.55 | 313.709 | 0.865 |
| | Deaths | Both | (0,1,0) | 183.884 | 185.381 | 0.09 |
| | | Female | (0,1,0) | 161.856 | 163.352 | 0.373 |
| | | Male | (0,1,0) | 170.094 | 171.59 | 0.241 |
| | DALYs | Both | (2,1,3) | 364.102 | 374.577 | 0.902 |
| | | Female | (0,1,1) | 335.295 | 339.784 | 0.158 |
| | | Male | (0,1,1) | 351.858 | 356.347 | 0.244 |
| Global | Incidence | Both | (1,2,0) | 67.603 | 70.534 | 0.948 |
| | | Female | (1,2,0) | 37.391 | 40.322 | 0.624 |
| | | Male | (1,2,0) | 51.503 | 54.434 | 0.83 |
| | Prevalence | Both | (2,1,1) | 282.333 | 288.319 | 0.951 |
| | | Female | (0,2,1) | 233.858 | 236.789 | 0.991 |
| | | Male | (1,1,2) | 263.855 | 269.841 | 0.896 |
| | Deaths | Both | (0,1,0) | 140.393 | 143.386 | 0.189 |
| | | Female | (0,1,0) | 110.762 | 113.755 | 0.968 |
| | | Male | (0,1,0) | 124.564 | 127.557 | 0.611 |
| | DALYs | Both | (0,1,0) | 341.897 | 344.89 | 0.223 |
| | | Female | (0,1,0) | 304.939 | 307.932 | 0.976 |
| | | Male | (0,1,0) | 323.772 | 326.765 | 0.671 |

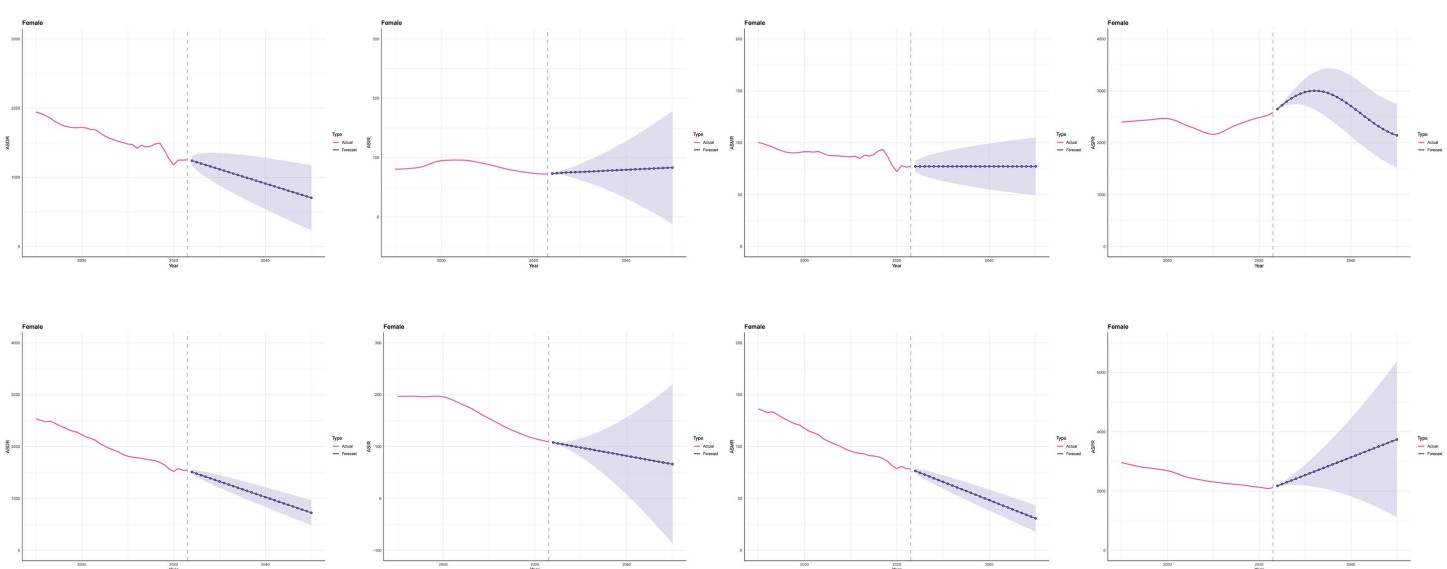

**Fig 8. Time trends of ASIR, ASPR, ASMR, ASDR in female IHD in China from 1990 to 2050. blue dot lines and shaded regions represent the predicted trend and its 95% CI.**

2,581.7 to a projected peak of 3,001.7 cases per 100,000 in 2032, before decreasing to 2,141.8 by 2050. Trends for ASMR and ASDR are consistent with those observed in the total Chinese population. The ASDR is forecasted to decrease from 1,254.3 to 704.1 cases per 100,000 people by 2050. In the global female population, ASIR, ASMR, and ASDR are expected to decline steadily, while ASPR is expected to continue rising. Specifically, ASIR is projected to decrease from 109.4 to 66.2 cases per 100,000, while ASPR is projected to increase from 2,116.3 to 3,740.4 cases per 100,000. ASMR is forecasted to decline from 78.3 to 30.8 cases per 100,000, and ASDR from 1,539.6 to 725.8 cases per 100,000 by 2050.

Fig 9 displays the projections for the Chinese male population. Both ASIR and ASPR are projected to rise until around 2025. ASIR is expected to increase from 110.3 to 113.3 cases per 100,000 people, peaking in 2036 at a projected 129.7 cases per 100,000. ASPR is projected to increase from 3,448.8 to 3,670.4, reaching a peak of 3,862.2 cases per 100,000 in 2028. ASMR is projected to remain stable over the forecast period. ASDR will exhibit an overall declining trend, falling from 2,133.3 to 1,635.1 cases per 100,000, despite temporary increases. Two local peaks in ASDR are projected for 2027 and 2031, reaching 2,107.2 and 2,107.5 cases per 100,000, respectively, followed by a subsequent decline to 2,013.5. In the global male population, the trends mirror those observed in females. ASIR is expected to decline from 177.7 to 105.6 cases per 100,000 people, ASPR will increase from 3,204.5 to 3,536.1 cases per 100,000, ASMR will decrease from 124.9 to 71.5 cases per 100,000, and ASDR will decline from 2,790.5 to 1,734.0 cases per 100,000 by 2050.

## Discussion

This study analyzed the trends in the burden of ischemic heart disease (IHD) in China and globally from 1990 to 2023. In 2023, the ASIR, ASMR, and ASDR of IHD in China's total population were lower than the global averages, while the ASPR was higher, which may be related to China's relatively low mortality rate. Notably, both ASIR and ASPR in China exhibited an overall upward trend during this period, in contrast to the global downward trend, suggesting that more robust IHD prevention and control measures are urgently needed in China to curb the growing epidemic. The decreasing trend in ASDR in China by 2023, compared to 2019 levels [12], may be attributed to improvements in the overall burden of disease, resulting from reduced premature death and disability, indicating progress in disease management and prevention strategies.

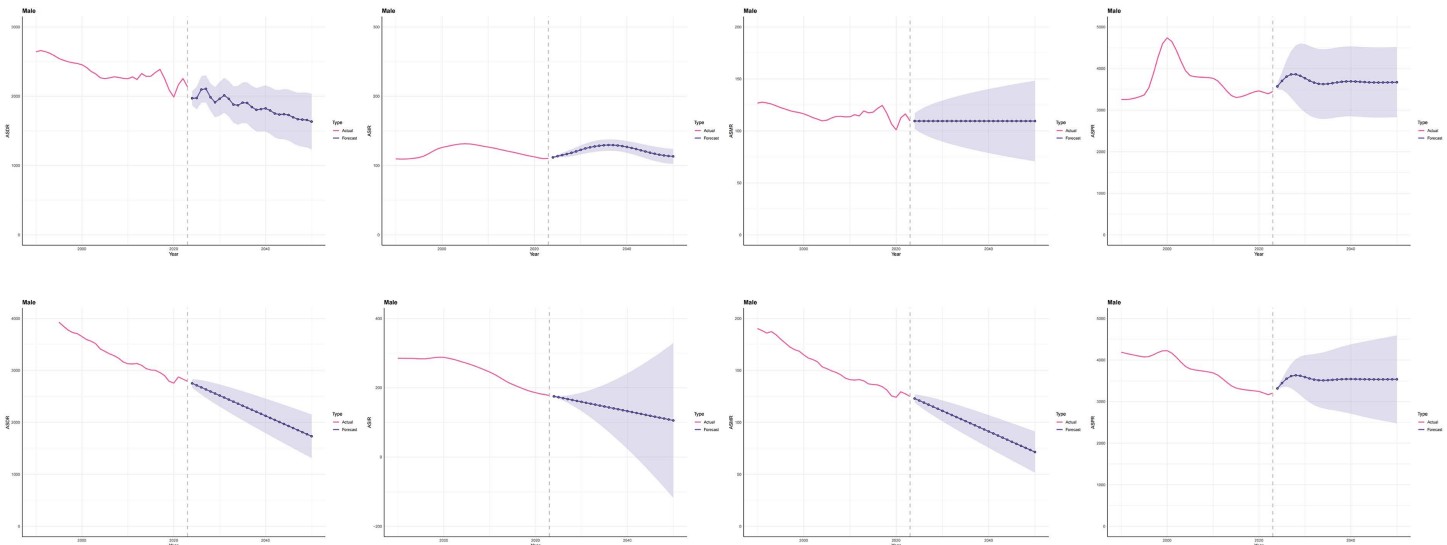

**Fig 9. Time trends of ASIR, ASPR, ASMR, ASDR in male IHD in China from 1990 to 2050. blue dot lines and shaded regions represent the predicted trend and its 95% CI.**

Two recent studies on the topic only analyzed data up to 2021 [13,14], whereas the present study extends the analysis period through 2023. It provides 2 additional years of data, which are particularly valuable for capturing potential shifts in disease burden following the COVID-19 pandemic, including recovery trajectories, healthcare system adaptations, and emerging post-pandemic health challenges. By including this extended period, our study offers a more current and policy-relevant perspective on the evolving global burden.

A more detailed age-group analysis revealed a fluctuating (tidal) pattern in the IHD burden across time in China. The decline in ASIR beginning in 2005 may be linked to the full-scale implementation of China's New Rural Cooperative Medical Scheme in 2003, which significantly expanded health insurance coverage and improved access to preventive healthcare services. As public awareness of health and preventive care increased, along with more frequent medical consultations, ASMR showed an initial upward trend. However, beginning in 2016, when percutaneous coronary intervention (PCI) technology matured and was widely promoted across China, both ASMR and ASDR began to decline. Meanwhile, ASPR increased between 2015 and 2017. A similar upward trend in ASIR, ASMR, and ASDR from 2020 to 2023 was observed globally, which may be attributed to the COVID-19 pandemic. The virus's cardiovascular complications, including acute myocardial injury, could have contributed to the temporary increase in IHD burden during that period.

When analyzing by sex, the ASIR, ASPR, ASMR, and ASDR were higher among males than females in both China and globally in 2023. Compared to 2019 [12], ASIR in Chinese males showed improvement. However, both male and female populations experienced a rise in ASPR, indicating that although China has made significant progress in IHD treatment—such as extending life expectancy and reducing mortality—there remains a substantial need for enhanced prevention and control strategies. Age-specific analysis further confirms that IHD morbidity and mortality remain predominantly concentrated in older age groups in both China and worldwide. The number of cases among individuals under 29 years old in 2023 was even lower than in 1990, whereas the proportion of cases occurring after the age of 55 was significantly higher. Therefore, targeted prevention and control efforts must continue to prioritize the elderly population. With the rapid aging of China's population [15], the number of IHD cases is expected to increase substantially in the coming years.

In summary, compared to the 2019 IHD data, China achieved notable progress between 2020 and 2023. However, efforts must now focus on the elderly male population. It includes strengthening IHD prevention policies, optimizing the allocation of health resources, and developing tailored public health interventions for elderly men with IHD. These strategies are essential and are expected to become the central focus of future prevention and control efforts.

Over the past three decades, population growth and aging have been major contributors to the sustained increase in the global burden of ischemic heart disease (IHD). In China, this burden has been primarily driven by a combination of both population aging and population growth, whereas globally, population growth has been the predominant factor. Studies have shown that the aging trend in China is particularly pronounced. It is projected that by 2050, individuals aged 65 and older will comprise nearly one-quarter of China's total population—approximately 400 million people—of whom 150 million will be aged 80 years or older [16]. Addressing the resulting pressure on the public healthcare system will be a major challenge for the nation.

We projected the burden of IHD through 2050 using the ARIMA model. The model predicts that China's age-standardized incidence rate (ASIR) and age-standardized prevalence rate (ASPR) of IHD will continue to increase through 2050, in contrast to a declining trend globally. Conversely, age-standardized mortality rate (ASMR) and age-standardized disability-adjusted life years (ASDR) are expected to decline in both China and the global population. However, the total population in China is projected to peak around 2030 and then begin to decline, whereas the ASIR among Chinese men is not expected to begin declining until after 2036. It underscores the importance of developing targeted, male-specific strategies to address the growing burden of IHD. Furthermore, existing research indicates that other modifiable risk factors, such as elevated blood glucose, dyslipidemia, hypertension, obesity, and smoking, are also significantly associated with IHD morbidity and mortality [17]. Lifestyle interventions, including weight management and smoking cessation, may therefore serve as effective strategies for reducing the IHD burden.

This study has several limitations. First, the Global Burden of Disease (GBD) estimates rely heavily on the availability and quality of underlying data. Limitations in health data collection and reporting systems, particularly in low-income countries, may lead to underdiagnosis and underreporting. Additionally, variations in diagnostic criteria across countries may introduce bias into the results. Second, due to the lack of province-level data within China, we were unable to conduct a more granular analysis of regional variations in the burden of IHD.

## Conclusions

Based on analysis of the 2023 GBD dataset, the prevalence of IHD in China remains somewhat inconsistent with global trends. Still, it is gradually converging toward the global average, particularly among elderly males. This pattern is projected to shift around 2030. China's rapidly aging population will place increasing strain on the national healthcare system, while population growth remains a major source of healthcare burden worldwide. To mitigate this burden, it is crucial to develop targeted prevention and treatment strategies, allocate healthcare resources effectively, and prioritize high-risk populations, especially elderly males. These actions will be crucial in mitigating the overall impact of IHD on public health in China and globally.

## Supporting information

**S1 Data. Data.**
(ZIP)

## Author contributions

**Data curation:** Zhuo Zhang, Guihua Yue, Longlong Wang.

**Formal analysis:** Zhuo Zhang, Guihua Yue, Longlong Wang.

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
