## [Decision Letter · Decision Letter 0]

3 Aug 2025

PONE-D-25-37188Analysis of Disease Burden and Future Trends of Ischemic Heart Disease in China and Globally, 1990-2023PLOS ONE

Dear Dr. Wang,

Thank you for submitting your manuscript to PLOS ONE. After careful consideration, we feel that it has merit but does not fully meet PLOS ONE’s publication criteria as it currently stands. Therefore, we invite you to submit a revised version of the manuscript that addresses the points raised during the review process.

Please make minor revision to the article to address reviewers’ comments.

We look forward to receiving your revised manuscript.

Kind regards,

Hean Teik Ong, FRCP, FACC

Academic Editor

PLOS ONE

Journal Requirements:

Additional Editor Comments :

Please make minor revision to the article to address reviewers’ comments.

Reviewers' comments:

Reviewer's Responses to Questions

**Comments to the Author**

1. Is the manuscript technically sound, and do the data support the conclusions?

Reviewer #1: Yes

Reviewer #2: Partly

2. Has the statistical analysis been performed appropriately and rigorously? 

Reviewer #1: N/A

Reviewer #2: I Don't Know

3. Have the authors made all data underlying the findings in their manuscript fully available?

Reviewer #1: Yes

Reviewer #2: Yes

4. Is the manuscript presented in an intelligible fashion and written in standard English?

Reviewer #1: Yes

Reviewer #2: No

5. Review Comments to the Author

Reviewer #1: The write-up provides a strong foundation for understanding the disease burden of ischemic heart disease (IHD) in China and globally. It is methodologically sound, using robust sources like GBD 2023 and appropriate tools like joinpoint regression and ARIMA modeling.

Reviewer #2: Major points:

The ischemic heart disease analysis have been completed by many different groups in various countries (PMID�40358980, PMID�40078456). So there is lack of novelty of this research.

Minor points:

The paper should be improved by native English speakers.

6. PLOS authors have the option to publish the peer review history of their article (what does this mean? ). If published, this will include your full peer review and any attached files.

**Do you want your identity to be public for this peer review?** For information about this choice, including consent withdrawal, please see our Privacy Policy .

Reviewer #1: No

Reviewer #2: No

---

## [Author Response · Author response to Decision Letter 1]

17 Sep 2025

Title: Analysis of Disease Burden and Future Trends of Ischemic Heart Disease in China and Globally, 1990-2023 Journal: PLOS One

Response to Reviewers’ comments

Dear Editor,

We thank the Editor for the careful consideration of our manuscript. We appreciate the comments, and we have made modifications to the manuscript. We carefully reviewed the comments made by the Reviewers and are providing our responses below. We hope that the Editor will find the revised paper suitable for publication, and we look forward to contributing to the Journal. Please do not hesitate to contact us with other questions or concerns regarding the manuscript.

Best regards,

Longlong Wang

E-mail: wanglonglong2022@stu.gxtcmu.edu.cn

Tel: +86-18677063120

Guihua Yue

E-mail: 54960313@qq.com

Reviewer #1

The write-up provides a strong foundation for understanding the disease burden of ischemic heart disease (IHD) in China and globally. It is methodologically sound, using robust sources like GBD 2023 and appropriate tools like joinpoint regression and ARIMA modeling.

Response: We thank the Reviewer for appreciating our efforts and our contribution to the literature.

Reviewer #2

The ischemic heart disease analysis have been completed by many different groups in various countries (PMID�40358980, PMID�40078456). So there is lack of novelty of this research.

Response: We sincerely thank the Reviewer for this insightful comment. We fully acknowledge that the topic may not be novel. However, in the field of public health, it is essential to continuously update existing knowledge to identify potential trends and emerging changes in a timely manner, which can then inform appropriate policies and interventions. The Global Burden of Disease (GBD) initiative is, by nature, a continuous and ongoing process that relies on the integration of updated data for meaningful interpretation.

Importantly, the two studies cited by the Reviewer only analyzed data up to 2021, whereas our study extends the analysis period through 2023. This provides two additional years of data, which are particularly valuable for capturing potential shifts in disease burden following the COVID-19 pandemic, including recovery trajectories, healthcare system adaptations, and emerging post-pandemic health challenges. By including this extended period, our study offers a more current and policy-relevant perspective on the evolving global burden.

Nevertheless, we added a statement about that in the Discussion.

The paper should be improved by native English speakers. Response: We thank the Reviewer. The manuscript was proofread.

Journal Requirements

1. Please ensure that your manuscript meets PLOS ONE’s style requirements, including those for file naming. The PLOS ONE style templates can be found at https://journals.plos.org/plosone/s/file?id=wjVg/PLOSOne_formatting_sample_main_body.pdf and https://journals.plos.org/plosone/s/file?id=ba62/PLOSOne_formatting_sample_title_authors_affiliations.pdf

Response: We have reviewed and ensured that our manuscript adheres to all of the style requirements outlined by PLOS ONE.

Response: We have provided the ORCID iD for the corresponding author and have successfully validated it in the Editorial Manager system.

Response: We updated the statement: “All data analyzed in this study are publicly available from the Global Burden of Disease project and are included in this published article.”

Response: We apologize for the discrepancy. We have corrected the grant information in both the ‘Funding Information’ and ‘Financial Disclosure’ sections of the submission system to ensure they are consistent and accurate.

Please upload the completed Content Permission Form or other proof of granted permissions as an “Other” file with your submission.

Response: Thank you for your guidance regarding the copyright of Figure 1. To ensure compliance with the Creative Commons Attribution License (CC BY 4.0), we have removed Figure 1 from the manuscript. The numbering of the subsequent figures has been adjusted accordingly.

Response: As suggested, we have reviewed the papers recommended by Reviewer #2. We found them to be highly relevant to our discussion, and we have therefore added them to the manuscript and the reference list.

Response: We have thoroughly reviewed our reference list to ensure it is complete and accurate. We can confirm that it does not contain any retracted publications.

---

## [Editor Report · Decision Letter 1]

22 Sep 2025

Analysis of Disease Burden and Future Trends of Ischemic Heart Disease in China and Globally, 1990-2023

PONE-D-25-37188R1

Dear Dr. Longlong Wang,

We’re pleased to inform you that your manuscript has been judged scientifically suitable for publication and will be formally accepted for publication once it meets all outstanding technical requirements.

As academic editor, I am happy to accept the article.  You still have to meet requirements of the administrative editors before the article can be published.

Kind regards,

Hean Teik Ong, FRCP, FACC

Academic Editor

PLOS ONE

Additional Editor Comments (optional):

As academic editor, I am happy to accept the article. You still have to meet requirements of the administrative editors before the article can be published.
---

## [Editor Report · Acceptance letter]

PONE-D-25-37188R1

PLOS ONE

Dear Dr. Wang,

I'm pleased to inform you that your manuscript has been deemed suitable for publication in PLOS ONE. Congratulations! Your manuscript is now being handed over to our production team.

Kind regards,

on behalf of

Dr. Hean Teik Ong

Academic Editor

PLOS ONE